# Physical Fitness as Part of the Health and Well-Being of Students Participating in Physical Education Lessons Indoors and Outdoors

**DOI:** 10.3390/ijerph17010309

**Published:** 2020-01-01

**Authors:** Marcin Pasek, Mirosława Szark-Eckardt, Barbara Wilk, Jolanta Zuzda, Hanna Żukowska, Monika Opanowska, Michalina Kuska, Remigiusz Dróżdż, Małgorzata Kuśmierczyk, Wojciech Sakłak, Ewa Kupcewicz

**Affiliations:** 1Faculty of Physical Culture, Gdansk University of Physical Education and Sport, 80-336 Gdansk, Poland; monkao84@wp.pl (M.O.); remik.pit@gmail.com (R.D.); wojciech.saklak@awfis.gda.pl (W.S.); 2Institute of Physical Education, Kazimierz Wielki University in Bydgoszcz, 85-064 Bydgoszcz, Poland; szark@ukw.edu.pl (M.S.-E.); zukowska@ukw.edu.pl (H.Ż.); michalinakuska@ukw.edu.pl (M.K.); 3Faculty of Public Health, Jozef Rusiecki University College, 11-041 Olsztyn, Poland; barbarawilk.awf@wp.pl (B.W.); malgorzatakusmierczyk@op.pl (M.K.); 4Faculty of Management, Bialystok University of Technology, 16-001 Kleosin, Poland; fitness2004@wp.pl; 5Faculty of Health Sciences, Collegium Medicum University of Warmia and Mazury in Olsztyn, 10-719 Olsztyn, Poland; ekupcewicz@wp.pl

**Keywords:** physical activity (PA) physical fitness, natural environment, outdoor and indoor physical education (PE) lessons

## Abstract

The analysis of existing information on physical activity and fitness as elements of health and well-being reveals that they are achieved particularly effectively in contact with nature. Physical education lessons outdoors, as a form of healthy training, have been performed in numerous countries for years, providing a response to the traditional indoor model of this kind of education. The purpose of this paper is to clarify the relationship between the participation of students in outdoor and indoor lesson activities and the change in their physical fitness. 220 students participated in an experimental study. The experimental group, which did exercise usually in open spaces, included 49 boys and 54 girls. The control group, which exercised inside school, consisted of 63 boys and 54 girls. The study period lasted two years and involved the fifth and sixth form of primary school. Experimental group subjects were 11.26 years old (±0.32) during the initial test, and the control group individuals were 11.28 years (±0.32). During the final test, the average ages of experimental group subjects was 12.96 years (±0.32), and 12.98 years (±0.32) in the control group. The International Physical Activity Test was applied in the study. The differences between the levels of particular components of physical fitness were not statistically significant during the initial measurement (*p*-values ranged from *p* = 0.340 to *p* = 0.884). After two years of outdoor physical education lessons, there was revealed a considerable increase in the speed, jumping ability, and aerobic endurance of the students. Statistically significant differences were observed in these three tests, including running speed (*p* = 0.001), legs power (*p* = 0.001), and endurance (*p* = 0.000). The findings encourage one to continue pedagogical experiments regarding physical activity in outdoor natural environments.

## 1. Introduction

The natural environment constitutes a collection of components which whether treated individually or as a cohesive whole, determine adequate recreational and sporting activities. Environmental conditions have a direct impact on general health [1,2] and human activity by developing energy costs of particular behaviours [3,4], and psychological balance [5,6]. 

The contemporary people during their existence evolutionally established relationships with wildlife [7]. Currently one may observe increasing distancing of the human being from nature of which he used to be an integral part, caused by the development of civilization. In consequence, a relatively small part of the human population uses health-improving values of nature to regenerate vital forces, including the filtration of contamination and noise, as well as the production of anti-microbial organic aerosols [8]. The trend of distancing from the natural world results in the resignation from natural environments as a place of recreation in favour of enclosed spaces, swimming pools, salt spa, sauna, body building gyms, bowling centers, billiards rooms or gymnasiums, which tend to be increasingly popular particularly in the fall-winter season.

These impacts may have stimulating or inhibitory effects, depending on the character of environment, the pace and directions of evolutions occurring in this environment. Nature may be treated as an intrinsic value, affecting the physiological and psychological state of an individual or as a basic value in use, without which the concrete forms of physical activity (PA) could not be executed. In the first depiction, which analyses the influence of nature on the possibilities of an individual’s functioning, there are implemented healthy, recreational [9], and cognitive-educational attitudes [10]. As regards the second depiction seen the possibility of the technical realization of physical activity (PA) in nature is conducive to developing sporting and recreational habits [11,12], although this relationship is not always noticed [13]. These behaviours may be developed due to numerous impacts of natural environment, ranging from the specific climate features to the elements accounting for geographical environment, such as relief, water, plants, and animals. Examples from other parts of the world prove that despite the increasing transformation of the natural environment due to human intervention, the awareness of the need to have a physical contact with nature is still high [14]. This is also achieved through physical activity (PA) in outdoor natural environments. Outdoor physical education (PE) classes, as a healthy training, are not new in many countries and provide an answer to the traditional indoor model of this education. School life exposes human body to long-term functioning in enclosed spaces at elevated temperatures, with a low level of humidity, artificial lighting, excessive dust, and often a much increased noise level, and it is the condition of air and improper thermal circumstances indoors that are considered to be the main limiting factors [15]. Therefore, the contact with an open space, for example, during a school break, provides an inspiration to avoid these limitations [16].

Important factors improving the quality of staying in nature were colours of nature, trees, aquatic areas, meadows, and diverse terrain relief providing opportunities for climbing, which as a whole, comprise a greater value than traditional playgrounds offered to these children [17]. Pupils who do exercises outdoors have enhanced their well-being in comparison with their counterparts exercising indoors [18] and are more eager to do exercises in mixed gender groups [19]. Sadly enough, over the recent decades, the opportunities for the development of children and adolescents in their contact with unspoiled nature have been gradually decreasing, which hinders their proper psychomotor development [20]. The health of the human body is increasingly evolving beyond its ecological conditioning, with a lack of awareness that nature comprises a nonorganic body of a human being and the human body is a part of the organic nature order [21].

In France there are conducted field based activities for secondary school students, including several disciplines, for example, climbing, canoeing, diving, skiing, or cycling tours [22]. In the United Kingdom, by contrast, outdoor activities are a compensation for the civilization development consequences and focus on learning about nature and its protection [23]. Germans, in turn, combine outdoor activities with school sport, tending towards creating the sports and recreational complexes with the possibility of being in a close contact with nature on the basis of theoretical cognition [24]. This issue has been noticed in Australia, where personnel training is organized to ensure the realization of outdoor education in the future. The training is of a theoretical and practical character, including weekend escapades, several day camps, as well as the so called TE, that is, an encounter with wildlife being a survival experience [25]. Likewise, in New Zealand there is a system of the teaching staff formation. Additionally, there are also easier forms of outdoors education, available for disabled children and adolescents, which are one of the best ways of personal development for their healthy friends participating in these forms [26]. 

Environmental education plays a special role in the United States. Many relevant programmes are supported by schools, educational centres, youth and religious organizations, as well as individual sponsors. Additional support is provided by scientific circles and the mass media. For this purpose, parks, forests, and dedicated zones in national parks have been made available. There are conducted for health educators [27] in these locations and the good American model ensures the realization of outdoor physical activities in the neighbouring areas of Canada on a similar basis [28].

In Poland these problems have been rarely noticed so far. As far as the organizational aspect is concerned, efforts are made towards the implementation of educational paths intended to the integration teaching in outdoor natural environments, combining the issues of biology, geography or history. In the school system of physical education (PE) it has been requested for a long time to create health paths, green outdoor gymnasiums, and to conduct some of PE classes outdoors in the surrounding area [29]. However, physical fitness was tested only in terms of endurance without taking other elements into consideration.

The organization of school physical education (PE) still faces a number of barriers. These include social conditions relating to pupils and organizational circumstances relating to school facilities. The current model of a sedentary lifestyle observed in the family proves that outdoor physical activity (PA) does not play a significant role among family members, or at least frequent contact with green space is not common. In addition, during their stay at school, children spend time in the school corridors or small rooms designed for corrective gymnastics. It happens quite often that a pupil enters the gymnasium for the first time only after the completion of the early school education. Then, physical education (PE) lessons are carried out in too numerous groups [30], sometimes even together with older pupils. The opinion that an escape from such a reality is only possible during short periods at the beginning and the end of school year, is shared still by too many teachers. The factors mentioned above, on one hand, are not conducive to shaping positive attitudes toward physical culture, including the attitude to PE lessons outdoors [31], on the other hand, when considering this phenomenon, these factors should make one develop an innovative formula of the child’s contact with nature. 

The purpose of the study was to clarify the relationship between the participation of students in lesson activities arranged in two ways and the change in their physical fitness. These two types of class activities should be understood as lessons conducted indoors and outdoors. This purpose has been determined in order to find out to what extent school activities carried out in various conditions cause different effects in terms of shaping adolescents’ attitudes to physical culture. Hence, the study attempted to review the effectiveness of implementing of school physical education (PE) in natural conditions. The changes in pro-somatic attitudes were compared in the experimental and control groups, taking the additional impact of individual and environmental conditions into consideration. In general, the objective was to determine whether pedagogical intervention in the form of an increased number of outdoor physical education (PE) lessons can bring about the expected changes in 8 analysed components of physical fitness. 

Therefore, the following question was defined: Do outdoor physical education (PE) lessons conducted over the period of two years have a larger impact on the changes in the individual components of physical fitness than PE lessons conducted indoors?

In reference to this question, the following research hypothesis was formulated: The two-year period of the experiment that consisted in conducting most physical education lessons outdoors suffices to determine an improvement in physical fitness against students from the control group.

## 2. Materials and Methods

### 2.1. Methodology

The project was implemented in the territory of Pomorskie Voivodeship. It was assumed that the children from experimental classes would participate in a significantly larger number of PE lessons outdoors than their peers from control groups. In our research, outdoor activities mean PE lessons conducted in sports school facilities in the vicinity of school buildings. The range regarding the variation of the number of PE classes was not strictly determined then initially, the previous research assumptions of Pańczyk [8] were taken into account; Pańczyk established the number of outdoor PE classes in experimental groups at the level of 75% and in control groups at 25–33%. In this study, in the experimental groups, there was finally achieved the average level of 60–65% and in the control groups 30%. The fact of the division into research groups did not mean that experimental group students exercised only outside and students from the control group – only inside.

In the Polish primary school system, a school terms lasts 18 weeks and there are four physical education lessons are held in one week. As a result, a maximum of 288 physical education lessons can be conducted during four terms. Therefore, students from the experimental group had 190 field lesson units on average, whereas control group students 85 lessons. Experimental group subjects had lessons inside the school when weather conditions were not suitable for exercises (rain, snow, strong wind, low temperature). The weather conditions in Poland during the recent years with relatively warm winters have resulted in only a couple of weeks during which it is more difficult to use outside sports facilities because of snow or low temperature.

The study participants performed the program in both groups without the need to diversify it for each of the groups. The core curriculum of general education for Physical Education subject in Polish primary schools defined by the Ministry of National Education indicates a uniform set of recommended physical activity in the fifth and sixth form. To a considerable extent, exercise can be performed both indoors as well as in outside school facilities. 

Bearing in mind that open spaces mean in our studies sports fields in the school area, there were no barriers to pursue the curriculum including outdoor and indoor lessons parallelly.

The exclusion criteria both for the experimental group and the control group were students who performed physical activity in sports clubs. The only possible extra form of exercise apart from physical education lessons included such non-school physical activities as cycling, rollerblading, and swimming lessons.

### 2.2. Participants

The study involved 4 schools in the northern part of the Pomeranian Voivodeship. In each school the project covered 2 groups of boys (49 boys in experimental and 63 in control group) and 2 groups of girls (54 girls in experimental and 54 in control group). Over the entire period of the project, the same students were involved in the investigation. The studies covered the period when the students were in the fifth and sixth form of primary school. The initial measurement was conducted at the beginning of school year in the fifth form (September). The final measurement was performed 21 months later, at the end of the sixth form (June). In the initial test experimental group students were 11.26 years old (±0.32), and control group students 11.28 years (±0.32) on average. During the final test, the average age in the experimental group was 12.96 years (±0.32), and 12.98 years (±0.32) in the control group. The research was finally completed by 220 students out of the initial number of 253 participants and only this group was analysed.

The initial and final tests compared the subjects from the experimental and control groups in terms of skinfold thickness. The measurement of fatty tissue thickness using the Lohman method [32] consisted in adding together the values of thickness of gastrocnemius muscle and arm triceps muscle skinfolds. The results are given in millimetres. We used a skinfold calliper of Fat Tester Accu-Measure Fitness 3000 (AccuFitness, LLC; Greenwood Village, CO 80155-4411; USA). In the initial test, students in the experimental group achieved 32.23 ± 13.32, and in the final test they had 31.76 ± 12.65. Students from the control group had 30.42 ± 12.75 in the initial test, and 30.80 ± 12.98 in the final one.

The substantive supervision over these classes was exercised by 10 teachers during the whole period of the project. In two out of eight cases, one teacher of a group changed after the first year of cooperation for reasons not related to this experiment. All teachers completed university education in the field of physical education, but they differed in gender, age, and seniority: among males, there was one person above 20 years old, two individuals were 15, one was 10, and one was 5. Among five females, there was one woman with job seniority of over 20 years and in case of the remaining four ladies it was about 5 years. They conducted classes in the schools covered by the study according to the same didactic-educational programme which contained all basic teaching contents and considered the opportunities for the programme implementation both outdoors and indoors. 

The physical activity curriculum in the fifth and sixth form assumes that the student:(1)performs and uses the following: bouncing the ball with a change of pace and direction, running and kicking the ball with a change of pace and direction, passing the ball with both hands and one hand while running, throwing the ball to the basket while running after dribbling (two-step shot), throwing and shooting the ball into the goal, passing the volleyball with both hands from the chest and bumping, doing three hits per side, doing a short serve, catching and throwing the ringo;(2)participates in small and simplified games;(3)participates in recreational games originating another European country;(4)organizes a selected sport or recreational activity in the peer group;(5)does other selected agility-acrobatic exercises (e.g., standing up on hands or head with assurance, doing a cartwheel);(6)does a set of agility-acrobatic exercises with or without equipment;(7)performs any jump over an obstacle with assurance;(8)throws a small ball running;(9)performs a long jump running and jumping over obstacles;(10)does part of a workout.

An initial and largely executed assumption was the fact that the lessons were carried out by the same teacher in the experimental and control group in every school. To remain impartial during measurements, the goal set for teachers was not connected with testing physical fitness, but only with the observation of social behaviour, such as team work and subjective assessment of well-being. Teachers did not know the physical fitness test results were a key element of the study. In addition to this measurement, they conducted many other assessments of physical fitness, which accounted for the final grade in physical education.

### 2.3. Research Method

The experimental study scheme was applied for the purpose of the study, and in this case it was an educational experiment. The suitability of this method with regard to pedagogical studies has been confirmed many times [33,34,35,36,37,38]. The method relies in investigating a phenomenon under normal conditions and acceptable manipulation involves the implementation of modified conditions by the researcher. The choice of the method required establishing the experimental and control group to ensure the demonstration of the range of changes occurring under the influence of a specific variable in the population involved in the experiment. The authors of the experiment assumed to align the experimental and control groups in such features as age, number of subjects, time of teaching and following instructions, skinfold thickness (differences in average thickness of skinfolds between the experimental and control groups were statistically insignificant). The experimental factor was the number of lessons conducted outdoors and was the only factor expected to differentiate both groups significantly. The principle, on which this study was based, was the so-called canon of a single difference by J.S. Mill [39] between the experimental and control groups, according to which the sequence in which the investigated phenomenon occurs, and an instance in which it does not occur, have every circumstance save one in common, that one occurring only in the former. The differing factor and the basic assumption of the experiment at the same time was the implementation of a different number of PE lessons outdoors in experimental and control groups. This factor could be defined as an experimental factor. Thus, this part of the procedure had a character of a diachronic analysis which ensured that a given number of individuals were subjected twice to the same measurements within a certain time frame, on account of a chosen set of characteristics, namely, the level of physical activity (PA) and knowledge, thus enabling one to determine not only if any changes took place, but also to identify the dynamics of their variability [40]. The study involved the classical model of concurrent groups and, after the initial measurement, an independent variable (experimental factor) was introduced in the experimental group, then, after less than two years (21 months) the measurement was repeated. In contrast, the experimental factor was not implemented in the control group after the first measurement, yet, the measurement was carried out again after some time. Then, the conclusions concerning the influence of the independent variable could be drawn on the basis of the differences between the results of the initial and final measurement. 

### 2.4. Research Tool

The research tool was International Physical Fitness Test [41] to measure general physical fitness and its individual components, carried out using stopwatches, measuring tapes, dynamometer, flags, poles, blocks and benches with cm graduation. The battery test consisted of 8 test components: 50 m run—test of running speed, standing long jump—test of leg power, endurance run—test of endurance, dynamometer handgrip strength measurement, straight-arm hang endurance—test of arm and shoulder strength, 4 × 10 m shuttle run test—test of maximal aerobic power, 30 sec. sit-and-reach test—test of abdominal muscles strength and forward bend—flexibility test.

The International Physical Fitness Test (IPFT) is a battery of tests developed with the cooperation between The United States Sports Academy and the Supreme Council for Youth and Sports [42]. The test was first introduced in 1977 as a two-day test battery, made up of the 50-m sprint, standing long jump, grip strength, 1000-m run, 30-sec sit-up, pull-up, 10-m shuttle run, and trunk flexion. 

Referring to the test, Polish authors developed fitness score which became the basis for test results. Despite the fact that, e.g., due to equipment limitations (lack of dynamometers in many schools), the test battery often consists of only five components, there are eight tests that are still popular, especially among Polish authors [43].

Although it is recommended to conduct the physical fitness test during two days, namely, test 1, 2 and 3 on the 1 day and test 4, 5, 6, 7, 8 – on the following day, it is acceptable to perform the whole test within one day with endurance measurement carried out at the end [41] and this model was applied in this study. Right before the test, the individuals were given appropriate instructions as to the way of performing the exercises. After a decent warm-up, pupils, dressed in sports outfit did exercises. All tests and measurements were conducted strictly in accordance with the instructions given beforehand [41,44]. All test results were converted into points and, with several options of score charts to choose from, the latest score chart was applied [41]. Theoretically, a student could reach up to 100 points from each test. Adding the points from particular tests, a summary score was obtained revealing the physical fitness level of individual pupils. Initial, periodical and final measurements were conducted and only the initial and final ones were used for the purpose of this study.

The decision to compare the physical fitness level in the control and experimental groups was based on the belief that physical fitness was connected with the range of pursued physical activity. Measurement by means of pedometer is a popular method to test the range of physical activity. Although the issue of physical activity using pedometers is common in the world literature, we have not found a reference of this problem to outdoor and indoor activities. Nevertheless, large-scale studies on the volume of physical activity outdoors and indoors were conducted in Poland [8]. The research carried out in the region of Zamość included as many as 2124 subjects from field groups made 7421 steps (±1361) on average, and students in the school gym – 6228 steps (±1080). Similar findings were observed in the research conducted in Kraków [45].

### 2.5. Statistical Analysis

The authors conducted a tool relevance analysis for the IPFT tool used in the study. In order to analyse relevance, the correlation analysis (r-Pearson) was carried out on standardised point results, which eliminated the need for a short time interval between repeated measurements. Very high rates were achieved ranging from 0.8766 to 0.988. Such an analysis was performed both in the experimental group and the control one, and the stability of results was almost identical. The reliability analysis was conducted using the Cronbach reliability coefficient, and assuming that test tasks reliably reflect the studied variable of “physical fitness.” Obtained reliability rates were very high – 0.9176 both in the first and second measurement.

The conducted test concerned objective changes as well as subjective indicators, that is why, to ensure their analysis there were applied methods evaluating these changes in terms of qualitative and quantitative data analysis methods. The research groups were divided according to research factors. Normal distributions of results were not identified in most cases, providing a basis to use non-parametric statistical significance tests. The Mann–Whitney U-Test was used to compare the experimental group with the control one, considering the changes taking place during two-year observations. The statistical significance level with a margin of random error 5% not exceeding for all calculations (*p* ≤ 0.05), highlight was assumed the statistically significant results.

### 2.6. Ethical Issues

The authorities of Gdansk University of Physical Education (PE) and Sport have been consulted on the testing proposals. The Ethical Committee, acting on the basis of the Regulation of the Minister of National Education and Sport of 9 April 2002 on the conditions for conducting innovative and experimental activities by public schools and institutions, concluded an agreement with the Pomeranian school superintendent concerning the project of evaluation and stimulation of the quality of physical education in (PE) schools in the Pomorskie Voivodeship (Project Number 17/03/05).

Permission to conduct the study in the schools was obtained from the individual head teachers. Before the survey, informed consent was obtained from children’s parents or guardians. Participants decided for themselves whether to participate or not. It was assured that participation was voluntary and refusal to participate in the study would not in any way affect adolescents in their studies or teachers in their work. Anonymity and confidentiality issues were emphasized throughout the study, as usernames were used rather than real names. Data were kept in a locked and safe place; only the researchers and the statistician were able to access the data.

## 3. Results

Due to the lack of normality in the distribution, the Wilcoxon signed-rank test was used for both groups to study differences between measurements. 

It was found that there was a statistically significant change in both groups in terms of performance of fitness tests. In order to determine the score level of the change in test performance, the change coefficient was also established expressing the difference in the final and initial results. 

As for the characteristics of the change in the results, there is a large spread of results between the min-max results within a group n, which confirms a much higher standard deviation. In the training group, lower results in the second measurement are only symbolic (hand grip and hang—a decrease by 1 point in one person); the subjects usually improved their results or remained the same level in individual tests. The total difference in IPFT results in control group tended both to get better and worse whereas, in experimental group the change only suggests an improvement in tests results (min > 0). The discrepancies shown in the characteristics were statistically significant in terms of all dimensions and the whole tool.

The analysis of individual elements of physical fitness of the experimental group and the control group during the initial measurement did not find any statistically significant differences. *P*-values ranged from *p* = 0.340 to *p* = 0.884. Detailed figures are presented in Table 1. Considering mean (M) results, the experimental group obtained slightly higher results in all tests. The greatest difference was observed in the hand grip dynamometer test (M_EG_ = 60.0; M_CG_ = 55.0), and the smallest difference was in endurance (M_EG_ = 60.0; M_CG_ = 59.0), arm and shoulder strength (M_EG_ = 60.0; M_CG_ = 59.0), agility (M_EG_ = 59.0; M_CG_ = 58.0), and abdominal muscles strength tests (M_EG_ = 59.0; M_CG_ = 58.0).

The research was repeated after two years of conducting the experiment. The results from the same subjects in the experimental and control group were used for an appropriate comparison (Table 2). Statistically significant differences were observed for three tests, including running speed (*p* = 0.001), legs power (*p* = 0.001), and endurance (*p* = 0.000). The results of these tests were significantly higher in the experimental group than in the control group. Students participating in outdoor activities also obtained a significantly higher overall result of the International Physical Activity (PA) Test (*p* = 0.003).

Differences in the level of specific components of physical fitness were not statistically significant in the initial measurement, yet such differences were observed during the final measurement (Table 3).

After two years of conducting physical education lessons outdoors there was shown a significant increase in the students’ speed, jumping ability and aerobic endurance in comparison with the group which usually did exercises indoors (Table 4). 

Statistical analysis included the results of individual tests during the first and second measurements in the control and experimental groups. No statistically significant differences were observed in the control group. However, students from the experimental group were reported to have significantly higher results in running speed (*p* = 0.0056), leg power (*p* = 0.0000), endurance (*p* = 0.0000), and agility (*p* = 0.0330) after two years.

## 4. Discussion

People as an integral part of life on earth, depend on the surrounding natural world. Although the current civilisation development even increases the distance from nature, it cannot completely isolate itself from the surrounding natural environment. 

Constant development leads to changes in the lifestyles of people today, turns an active lifestyle into a sedentary one, reduces the time spent in the open air, and offers increasingly processed, high-calorie, and widely available food. As Skłodowska reports, lack of exercise and staying indoors without the impact of natural light is thought to contribute to many of today’s illnesses (e.g., heart diseases, obesity, joint and spine ache) [46]. Kwilecka, in turn, claims that exercise done in direct contact with nature, sounds, fresh air, and influence of the sun and oxygen, has a special health value [47].

Based on the authors’ experience, when analysing the course of physical education (PE) lessons, it was resolved to conduct an experiment to define the impact of these outdoor lessons on the level of students’ physical fitness and endurance, and thus on their health. Osiński reports that cardiac-respiratory efficiency, strength, muscle endurance, flexibility, and fatness are generally considered to be crucial in the optimal health [48]. As regards the work of a physical education (PE) teacher, the most important positive health measures include physical development, physical activity, fitness, and endurance. Therefore, these elements are most often tested to providing an overall diagnosis of the functioning of a child’s body [49].

Physical activity with access to green space may enhance physical fitness more efficiently than activity performed indoors, commonly understood as traditional teaching. Clear differences can be observed here in favour of outdoor activity in terms of balance and coordination [50,51]. The quoted authors add that thanks to positive results of outdoor activity, green space near school plays a vital role in promoting physical activity, emphasizing that still not enough correlations of this type have been revealed though [7]. Social competence and, consequently, social capital can be shaped through outdoor education. Interesting as it appears to be is the fact that students with worse results, due to lower mobility, prefer to stay indoors as moving in open space requires greater engagement from them and is simply more difficult [52]. 

Experimental study conducted in Poland and focusing on the comparison of effects of PE lessons in students aged 13–15 revealed better effects of outdoor activities within such aspects as the number of movements, heartbeat frequency, emotional feelings, attitude to physical culture, and endurance [29]. Own studies also supported the idea that physical activity in the open air has a positive impact on physical fitness level of the subjects. Both groups (experimental and control), after two years of experiment, improved the results achieved in fitness tests, which may be a normal sign of growing up; however, the group that exercised more outdoors (experimental) obtained a much higher increase in results, and in four tests (running speed, leg power, aerobic endurance, maximal aerobic power), this improvement was statistically significant. The control group that did more exercise in closed spaces did not achieve a statistically significant improvement in physical fitness test results.

Both laboratory and outdoor measurements of physical activity using pedometers revealed a high level of physical activity with oxygen consumption (r = 0.62–0.93) and with direct observation of the tiredness level (r = 0.80–0.97) [53]. Echocardiograms confirmed the differences in the intensity of activities conducted outdoors and indoors [8]. This was proved by mean heart rate values amounting to 159 beats per minute outdoors in relation to 142 beats in the gym. Significant differences were shown from the seventeenth minute of a lesson in the statistical significance levels calculated for all minutes during a lesson. 

The students’ endurance was assessed by means of the Cooper test carried out four times during the so called Zamość test [8]. Measurements performed in subsequent tests proved bigger progression of meters covered in experimental groups, however the differences were not statistically significant in most cases. 

Function tests were a reference to the above-mentioned assessment. As evidenced much earlier, a significant correlation between heart rate and the frequency of stepping up and down, ensuring the determination of the level of aerobic endurance, was also observed during the test in the population of Zamość [8]. Stepping up and down 33 times at a varied height, depending on the length of lower limb, is the standard used regarding the n-170 rate to achieve a heart rate equal to 170 [8]. Children in Zamość [8] achieved better results, which turned out to be significantly better in the groups that had most PE lessons in the natural environment. Own research revealed that children who had outdoor PE lessons obtained statistically better results in the assessment of endurance, which is essential for respiratory and blood circulation systems.

The capacity of physical activity (PA) measured by the number of moves per time unit revealed bigger activity among the individuals pursuing outdoor physical activities [45]. The obtained results also demonstrate a positive impact outdoor activities on the speed of movement (50-m run), speed and change of direction (agility—4 × 10 m shuttle run test carrying a block), or the speed of rising from lying to sitting (assessment of abdominal muscles strength). The study conducted in Zamość showed the capacity of work expressed by the number of arm and leg movements during physical education classes outdoors was about 20% bigger than indoors [8]. This was also true with preschool groups from Sweden and the United States [54], and in case of British research, which revealed an approximately 2.5 times bigger physical activity difference in children doing exercises outdoors in comparison to the indoor group [55]. Besides, no difference was shown between boys and girls when regards the time spent in the open air, and outdoor physical activity was reported to be higher in the summer than in the winter, yet this seasonal character of differences was not observed in case of indoor physical activity [55]. Likewise, the American studies conducted among preschool children revealed that physical activity conducted outdoors improved the frequency of this activity, which was confirmed by an increased number of movements more time engaged in physical activity [56]. The students involved in the experiment (participants of outdoor activities) conducted by the authors, obtained a significantly higher overall result of the International Physical Activity Test, proving better overall physical fitness. This means that the subjects achieved better results in all fitness tests. As mentioned before, components of physical fitness, as strength, speed, flexibility, and endurance, comprise elements of optimal health.

Still other studies suggested accelerated biological development during physical activity in natural environments [57] and increased mobility of children having physical education outdoors, despite the fact that these classes lasted for a shorter period of time [58]. As far as the Norwegian studies are concerned, an experimental group did exercises in the forest near the kindergarten for two hours every day over one year, while the control group spent time in immediate neighbourhood of the kindergarten at the same time. However, the findings of both groups were not compared and only pre-test and post-tests results were assessed separately for each group. The control group improved its results in three out of nine fitness tests whereas the experimental group eight tests [59]. Own research carried out over two years revealed that the experimental group, that participated in outdoor activities at the level of 60–65%, had a significant improvement in all fitness tests, whereas the control group that took part in outdoor PE classes at the level of 30%, improved their results to a small extent. In the first group participating in activities in the open air, the difference between the first and second test in the assessment of overall fitness (an average score of overall IPFT) was 29 points, while the control group that had more indoor classes, scored only 5 points.

A positive impact of natural environment elements on the human does not depend on age and does not relate only to physical fitness, but to the health condition in general. Improved health condition was observed in elderly people who had physical activities outdoors in comparison to those doing exercises indoors. The data gathered from six cities suggested also a worse health condition in 6.9% of subjects doing exercises indoors and only in 3.4% of the outdoor subjects. A good health condition was, in turn, identified in 58.2% of indoor respondents and as many as 72.4% of their outdoor counterparts [60]. 

Therefore, returning to the natural environment, spending time outside, undertaking spontaneous exercise, and above all, skillful arrangement of physical education (PE) lessons, appear to be obligatory for each PE teacher. Exercise in the fresh air, in different weather and site conditions, at the playground or outdoor gym has a positive influence not only on health, but also cognitive processes of people at different age. Exercise as a drive to enhance fitness and endurance should be used a preventive measure.

## 5. Conclusions

A comparative analysis of own results obtained in the initial test revealed no differences between students selected for the experimental group and the control group. This similarity was observed with regard to the effects of individual physical fitness tests. The final study revealed the variation of outcomes between both populations in favour of the experimental group. 

Based on the studies, the following conclusions have been drawn:students who participated in PE classes that were usually conducted in the open air were distinguished by better physical fitness than their peers doing outdoor exercise to a smaller extent;undertaking activity in the open air contributed to significant improvement in the legs power and speed in the case of students taking part in the two-year experiment;the most clear changes in physical fitness of the subjects were observed in the case of endurance, which suggests the stimulating influence of outdoor PE lessons on this aspect of their fitness;outdoor physical education classes should prepare students for a skilful use of natural sites in their intermediate surroundings to take up physical activity. Physical education teachers should encourage and prepare students for lifelong physical activity by providing the ability to take part in systematic and active leisure in the open air, which is conducive to developing physical fitness and endurance, and thus health.

The above presented findings encourage one to continue pedagogical experiments regarding physical activity in natural environments. They may turn out to be useful to confirm the results published so far and to provide opportunities to broaden the knowledge of shaping health and physical well-being through contact with natural surroundings. At a time, where stress, lack of time, automation, and mechanization dominate, the meaning of such discoveries will become increasingly important. 

## Figures and Tables

**Table 1 ijerph-17-00309-t001:** Comparison of physical fitness of the students in the experimental group in the initial and final measurement and the control group in the initial and final measurement. The T score formula enables to take an individual score and transform it into a standardized form, one which helps to compare scores. Z value refers to a standardized score, also known as a z-score.

Test	Experimental Group (*n* = 103)				Control Group (*n* = 117)			
	N	T	Z	*p*-Value	N	T	Z	*p*-Value
Running speed	103	0.000	8.810402	0.000 *	72	409.000	5.078583	0.000 *
Leg power	103	0.000	8.810402	0.000 *	72	698.000	3.456803	0.000 *
Aerobic endurance	103	0.000	8.810402	0.000 *	69	652.000	3.321305	0.000 *
Hand grip strength	77	10.500	7.570503	0.000 *	59	330.500	4.185353	0.001 *
Straight arm hang endurance	86	18.500	7.974706	0.000 *	68	661.500	3.125446	0.000 *
Maximal aerobic power	82	0.000	7.865922	0.000 *	53	267.500	3.966039	0.000 *
Abdominal muscles strength	86	0.000	8.054367	0.000 *	71	661.500	3.532444	0.000 *
Flexibility	79	0.000	7.721570	0.000 *	51	217.500	4.175879	0.000 *
Total	103	0.000	8.810402	0.000 *	110	1311.500	5.192188	0.000 *

* *p* ≤ 0.05.

**Table 2 ijerph-17-00309-t002:** Differences between the results of the final and the initial measurement in the experimental and control groups.

Test	Experimental Group (*n* = 103)					Control Group (*n* = 117)					*p*-Value
	Mean	SD	Min	M	Max	Mean	SD	Min	M	Max	
Running speed	5.91	1.70	3	6	11	1.08	2.24	−5	0	10	0.000 *
Leg power	5.76	1.89	2	6	15	1.05	3.05	−6	0	11	0.000 *
Aerobic endurance	10.65	3.52	5	10	20	0.86	2.89	−6	0	19	0.000 *
Hand grip strength	1.59	1.34	−1	2	6	0.91	2.18	−4	0	9	0.000 *
Straight arm hang endurance	1.64	1.27	−1	1	7	0.95	3.17	−8	0	21	0.000 *
Maximal aerobic power	1.58	1.23	0	2	6	0.91	2.26	−3	0	11	0.000 *
Abdominal muscles strength	1.60	1.10	0	1	4	1.00	3.38	−10	0	16	0.000 *
Flexibility	1.47	1.19	0	1	5	0.83	3.23	−23	0	9	0.000 *
Total	30.20	8.53	11	29	52	7.11	13.28	−34	0	47	0.000 *

* *p* ≤ 0.05.

**Table 3 ijerph-17-00309-t003:** Comparison of physical fitness of the students in the experimental group and the control group in the initial measurement.

Test	Experimental Group (*n* = 103)					Control Group (*n* = 117)					*p*-Value
	Mean	SD	Min	M	Max	Mean	SD	Min	M	Max	
Running speed	60.6	11.5	33	62.0	86	59.5	13.5	30	58.0	89	0.506
Leg power	59.3	10.2	37	60.0	89	59.4	10.0	34	58.0	84	0.837
Aerobic endurance	58.8	10.9	29	60.0	84	59.7	11.3	28	59.0	95	0.860
Hand grip strength	57.6	10.3	33	60.0	81	57.4	10.9	30	55.0	81	0.587
Straight arm hang endurance	58.3	12.0	0	58.0	86	57.3	12.7	0	57.0	85	0.396
Maximal aerobic power	59.5	11.8	1	59.0	88	57.9	12.6	0	58.0	85	0.340
Abdominal muscles strength	58.5	10.5	23	59.0	81	59.3	10.9	33	58.0	85	0.884
Flexibility	60.5	11.1	31	62.0	82	59.3	11.8	31	60.0	87	0.272
Total	473.3	70.8	287	478.0	660	470.1	74.3	222	465.0	653	0.385

**Table 4 ijerph-17-00309-t004:** Comparison of physical fitness of the students in the experimental group and the control group in the final measurement.

Test	Experimental Group (*n* = 103)					Control Group (*n* = 117)					*p*-Value
	Mean	SD	Min	M	Max	Mean	SD	Min	M	Max	
Running speed	66.5	10.9	40	66.0	90	60.6	13.2	29	60.0	89	0.001 *
Leg power	65.1	10.1	43	66.0	93	60.4	10.6	30	60.0	86	0.001 *
Aerobic endurance	69.4	10.6	40	70.0	92	60.5	11.9	25	60.0	93	0.000 *
Hand grip strength	59.2	10.2	37	60.0	83	58.3	10.8	30	57.0	86	0.385
Straight arm hang endurance	60.0	11.5	7	60.0	89	58.2	12.2	10	59.0	86	0.189
Maximal aerobic power	61.1	11.5	5	62.0	90	58.8	12.9	0	59.0	86	0.153
Abdominal muscles strength	60.1	10.4	24	60.0	83	60.3	11.3	33	59.0	87	0.827
Flexibility	62.0	10.9	34	64.0	86	60.1	11.9	29	62.0	87	0.166
Total	503.5	68.8	324	507.0	693	477.2	79.4	205	470.0	677	0.003 *

* *p* ≤ 0.05.

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
