# Peer review of "Physical Fitness as Part of the Health and Well-Being of Students Participating in Physical Education Lessons Indoors and Outdoors"

_ijerph, 2020, doi:10.3390/ijerph17010309_

Round 1

Reviewer 1 Report

The revised manuscript has been improved. It seems that the Authors considered reviewers' comments and suggestions.

However, Authors should pay more attention to details. Please, use only the abbreviation in the main text (PA, PE), while you have introduced them in the abstract. For bibliographic references, criteria must be unified. One more time, double-check the References section and follow the IJERPH Journal style. Also, I have noticed some repetition in the text, for example, page 2 (line 44) and page 8 (line 340) “The contemporary man during his existence cannot become completely isolated from the natural environment due to evolutionally established relationships with wildlife [7]”. Please, be aware of that and try to avoid them. Please, unify the font in the manuscript (for example, page 6, lines 262-265).

Additionally, it will be good to justify the time of the natural experiment. Why did Authors choose 21 months (almost two years)? Any references for that?

In the case of accepting the manuscript, I would suggest the Authors discuss the issue of the limitations and strengths of the study and also try to include mention of the originality of this study (Discussion section).       

Reviewer 2 Report

Review of the manuscript entitled "Physical fitness as part of the health and well-being of students participating in physical education lessons indoors and outdoors ".

After the modifications and improvements made by the authors, the manuscript is proposed for publication.
Only bibliographical references that are not in the appropriate format (Vancouver) should be reviewed.

Reviewer 3 Report

The authors have made substantial changes to the manuscript, but some of the fundamental errors relating to study design and statistical analysis remain.

These are:

It's still unclear what happened indoors and outdoors and any important differences such as amount of space.  This should be clear from Section 2.3, but the essential details have not been provided.  Lines 235-236 states that "number of lessons indoors and number of lessons outdoors was the only factor expected to differentiate both groups", but it doesn't seem to be something that was checked.  It seems difficult to believe that the same lessons would take place indoors and outdoors e.g. there is often more space outdoors and teachers might take advantage of the greater space for more running and speed type activities - i.e. activities that are relevant to the research.  Likewise, indoors is often preferred for gymnastics type activities.  I couldn't see anything in the design that allowed for observation of the activities that occurred, but perhaps there are teacher records or other data available that could be used to check the activities.

Details such as the reliability and validity of the IPFT are not provided.  There is one reference to a 1974 report and a more recent reference that reports use of the test but does not report psychometric data.

A repeated measures design has been used, so statistical analyses appropriate for repeated measures need to be used.

Other points:

The experiment is still referred to as a 'natural experiment', but it doesn't meet criteria for a natural experiment.

The new introduction has gendered language that is not appropriate for contemporary academic writing (line 44).

Lines 44-53 increase the confusion about key terms/constructs, such as nature and outdoors.  If the manuscript is revised, it needs to be very clear what is meant by indoors and what is meant by outdoors in this study.  

Round 2

Reviewer 3 Report

I appreciate the difficulties of revising analyses when the statistician has retired, but it doesn't seem to be a valid reason to leave the analysis uncorrected.  This is a repeated measures design and it is essential to use statistical analysis for repeated measures.

I checked the chapter the authors refer to regarding the definition of natural experiments.  DiNardo refers to these as 'serendipitous situations'.  The experiment conducted in this study was planned and should not be referred to as a natural experiment.

I couldn't access the articles mentioned regarding development and use of the IPFT.  The response of the authors doesn't mention reliability or validity of the test though, and that's what my query was about.  If a test has poor test-retest reliability, for example, it is not a good test to use for repeated measures design.  I assume reliability and validity have not been checked.  I appreciate that other metrics have been established, but these do not replace the need to understand the reliability and validity of the IPFT.

I'm still unclear on what has happened indoors and outdoors.  I can see in the revised version that it is clearer that it isn't about what happened, but about the number of lessons occurring indoors or outdoors. 

Author Response

We thank for the Reviewer's comments. Please find our answers in the attachment.

This manuscript is a resubmission of an earlier submission. The following is a list of the peer review reports and author responses from that submission.

Round 1

Reviewer 1 Report

The authors present the results of a very interesting research project, examining the physical adaptation of students who do more physical education in the environment in comparison to indoors.  Overall the design of the study was clear and the work presented as experimental research.

Introduction

The literature about this topic was well presented and provided some background. However many more studies were included in the Discussion which I suggest detracted from the Discussion itself. I recommend reviewing all literature presented in the entirely of this manuscript and rework the Introduction.  Suggest also include only that literature that pertains to the age group that the research itself focuses on.

Line 35 – ‘psychological balance’ – it is not clear what this means. Please clarify and rework.

Line 57 - Rework. It is unclear what is meant by ‘Significant nature assets…..’

Lines 121- 122 – Suggest that this last paragraph is problematic. It is unclear what is meant by lines 120 – 121. Also why not include a hypothesis, the project is clearly experimental?

Methodology

More information is required about the age range of the participants.  Would also suggest that some more detail about the actual physical education programmes could be included. It is difficult to see what the actual activities were.  Were these predominately aerobic? The results suggest this, however it is unclear. Suggest that the considerably more detail is required about what the groups actually did.

A lot of work has gone into presenting the experimental design. A hypothesis should follow this. 

Statistical Analysis

What did the test of normality actually reveal? Some detail about the analysis is needed.

Results

The comparisons between the groups is clearly presented. Suggest the results from ‘within the groups’ could have been shown as well. Some Figures may have also been helpful, particularly for the ‘Total’ scores.  

‘Legs power’ should be ‘Leg power’

Discussion

The Discussion needs a complete rewrite.  The actual results of the research are not discussed at all in this section. It does include a large amount of literature about the topic which as said above should be in the introduction. The Discussion should focus on the following.

What were the significant findings from the research? Some full explanations about what was found needs to be presented. I suggest a focus on the predominance of improvement in the aerobic aspects. Some comparisons with similar literature to highlight the findings by comparison. New literature is in the Discussion and should have been presented in the Introduction and then mentioned again if the conclusions were similar or not as the case may be.

There needs to be some explanation offered about variables that did not change. These are finding as well. This was a 2 year study so no change in flexibly could also have some explanation. This highlights the point above about what activities the students were actually undertaking.  The results of the strength assessment also needs explanation.

Some information about the implications of this work in terms of practice also needs emphasis. What are your recommendation about students and participation in physical activity? What do you recommend for future research?

Overall the Discussion and Conclusion need to be rewritten to focus more fully on the actual results of this study. I suggest the changes ‘within groups’ could be looked at firstly and then the comparison between the experimental and control.  The mechanisms of physical adaptation to exercise should be a focus, such as to why some variables change and others do not. This needs to be clear as one might assume that the changes were due to growth, development and physical maturation over 2 years, rather than being in the outdoors. Results and Discussion need a full rethink.

Reviewer 2 Report

I applaud the authors' attempt to examine the relationship between physical education settings (indoor and outdoor) and levels of health-related fitness. As a reader, there are several major concerns that need to be considered and perhaps expanded upon.

1st: there needs to be more demographic information provided pertaining to the participants. If the topic of discussion is health-related fitness measures, then reporting the participant’s BMI, etc. is important. From a motor development perspective, the influence of growth and maturation should be noted. It is not a fair assessment to just say the experimental group had greater improvements after two years without examining the impact on physiological factors of children during that phase in their life.

2nd: The authors claim that outdoor activities have a greater impact on a student’s health; however, there is no description of the activities completed during these two years of physical education. The difference in fitness levels could be attributed to numerous other factors. To be able to claim that having students outdoors impacts health more than indoor classes the authors need to provide details regarding the types of content taught and the pedagogical strategies employed. If both settings taught the same content with the same lesson objectives then yes you have cause to make those claims; however, if the two settings were taught using different teaching styles, curriculum and standards then as a reader I find little value in those claims. Additional information related to the number of classes during those two years is needed. Where the outdoor classes all outside? Meaning when there was bad weather did those students also engage in indoor activities. These are all confounding factors that need to be mentioned.

The research cited in this article mainly focuses on physical activity and the benefits of outdoor space and activities, if this is the case then yes having more wide-open space for a child to run would elicit the opportunity for more movement but the authors speak to physical education. Physical activity is a product of a quality physical education class and if the authors are using physical education and physical activity interchangeably then they have a stronger basis for their claims. If the claim is that an outdoor physical education setting elicits more physical activity and the link between PA and fitness is greater than that student’s in an indoor setting, then as a reader I would ask for evidence of a PA measure. Saying the students did better fitness-wise does not offer the reader evidence of why.

The results presented in this study are minimal. The readers are provided with a sentence and a table, to sum up, the findings of the study. More information and further analysis are needed in order to draw more conclusions. 

Reviewer 3 Report

Review of the manuscript entitled "Physical fitness as part of the health and well-being of students participating in physical education lessons indoors and outdoors".

The manuscript submitted is appropriate to the subject matter and scientific rigor.

After an exhaustive revision of the text, it is considered that the authors should update the bibliographical references given that the most current one is from 2015. There are many recent and relevant studies that address the subject matter of the manuscript.
In addition, the authors should clarify the styles of pedagogical intervention used in research.

Reviewer 4 Report

It's great to see a study examining differences in benefits achieved in PE indoors and outdoors.  Unfortunately though, there are a lot of problems with the present manuscript.

I found this article difficult to read for three reasons.  First, the grammatical errors are such that it is often difficult to understand the sentences.  Second, there are a lot of country specific terms relating to the schools and class years (e.g. second educational stage - perhaps the main point is the age of the participants??).  Third, the paper doesn't seem to follow conventions such as reporting mean age of participants in each group.  If there are differences in age, it is possible that one group will show greater gains based on developmental trajectory alone.

I found it hard to understand what happened over the two-year 'intervention' period.  It's just described as indoor vs outdoor, but were children receiving the same instruction in these two contexts?  It would seem unusual if this was the case as it means the teachers were not taking advantage of the different opportunities in each context (e.g. there aren't many opportunities to run in snow when indoors).

Outdoors is assumed to be 'nature' but we don't have a description that demonstrates what is meant by nature.  A basketball court is outdoors, but wouldn't qualify as nature, for example.  This is particularly important because the discussion has a very strong nature focus, but we don't know what nature was available in these classes.  We also don't know e.g. if participants had more square metres of space outdoors, wore different clothing, had differently qualified instructors.  I acknowledge that it is impossible to keep everything equal other than the independent variable of interest in these types of studies, but there should still be an attempt to examine the impact by e.g. including as a covariate in the analysis.

The authors seem to have misinterpreted what is meant by a natural experiment.  A natural experiment occurs when a decision external to the researcher is made to create two or more groups with different conditions.  It looks as though the researchers allocated children to different conditions (based on information on lines 161-164).

The IPAT seems to be the only measure used.  There are no psychometrics reported in the paper.  Reference is to an article from 1996 and one from 2005 and neither are in English which will make it difficult for readers to understand if the tests were appropriate.  The convention is to provide the psychometrics and I would also recommend including some recent publications in English if possible. 

I'm not sure why a nonparametric test was selected, it doesn't seem to be mentioned.  The Mann-Whitney U-Test isn't a repeated measures test.  A repeated measures test is needed for this design and effect sizes should be reported.

There is a lot of terminology and changes in direction.  For example, line 114 introduces the term 'pro-somatic attitudes' - which is never defined, but is argued to be measured as a difference between experimental and control (although I can't see how that was done).

I'm also unsure about the research question on lines 123-125.  It's argued that there is a lack of research to help formulate a hypothesis so an exploratory question is offered.  There is actually a lot of research that could be used to generate hypotheses.

It's not clear who administered the IPAT. Was the person blinded to the aims of the study?  Is there potential for experimenter bias?

There are other aspects of the manuscript that should have been fixed before submission e.g. the abstract refers to two instruments, but then only names the IPAT.  There are similar inconsistencies throughout. 

It is possible that with a major revision the paper could be accepted.  I suspect though that it will be difficult to get this article to IJERPH level.  It looks as though participants received different conditions for their PE and there was potentially a lot of variation, not just indoors/outdoors.  The only measure is the IPAT and it is of unknown psychometric properties and was potentially administered by researchers who were not blind to the aims of the study.

Reviewer 5 Report

Page 1, lines 22-23: Please, add more detailed information about your research group (for example the age of the participants, the number of girls and boys).

Page 1, lines 24-25: If there were two measuring instruments, then please add the name of the second one.

Page 1, lines 25-28: Please, add some statistical information about the results. Introduction/Methodology/Discussion/Conclusions: Please, add abbreviations for physical activity (PA) and physical education (PE) in the manuscript.         

Page 2, line 91: Please, add the “Physical Education” before the abbreviation - “PE”.

Page 3, line 126: I suggest to change the name of the section “Materials and Methods” to “Participants and Methods”, while your research group is a group of people.

Page 3, line 141-142: 220 students participated in the study. Please, add more detailed information about your research group (for example the mean age of the participants, the exact number of the participants and the number of girls and boys).

Page 3, line 141: “The study period covered the years which they had second education second stage, including forms 4-6”. The 4-6 forms are not clear form me – please try to clarify.

Page 4, lines 156-157: “The suitability of this method with regard to pedagogical studies has been confirmed many times” – it requires more references.

Page 4, line 177: “The measurement was repeated after a while” – You have to be more specific.

Page 4, line 183: Please, add an appropriate reference to the “International Physical Fitness Test”.

Page 6, Results: The description of the results is quite poor. Please, add some statistical information to the results. Also, all tables should be unified – please, try to avoid horizontal lines.

Page 7, line 262: “…the weaker students” – please, try to replace this term for a different one -  more appropriate one.

Page 7, line 265: “Experimental studies…” means more than one, therefore, add more references here.

Page 8, line 315: “The Turkish studies…” – more than one? Then, please, add more references here.

Page 8, Discussion: Please, indicate some limitations and strengths of the study and also try to include mention of the originality of this study. It is recommended to elaborate and emphasize some potentially novel aspects presented here.

Page 8, Conclusions: Authors should mention the value and some practical applications of this study.

Pages 9-11: For bibliographic references, criteria must be unified. Please, double-check the References section and follow the IJERPH Journal style. Also, all references should be in English.
